# Typical and Atypical Symptoms of Petrous Apex Cholesterol Granuloma: Association with Radiological Findings

**DOI:** 10.3390/jcm11154297

**Published:** 2022-07-24

**Authors:** Alessandro Vinciguerra, Mario Turri-Zanoni, Benjamin Verillaud, Jean-Pierre Guichard, Luca Spirito, Apostolos Karligkiotis, Paolo Castelnuovo, Philippe Herman

**Affiliations:** 1Otorhinolaryngology and Skull Base Center, AP-HP, Hospital Lariboisière, 75010 Paris, France; benjamin.verillaud@aphp.fr (B.V.); philippe.herman@aphp.fr (P.H.); 2Unit of Otorhinolaryngology, Department of Biotechnology and Life Sciences, University of Insubria, Ospedale di Circolo e Fondazione Macchi, Via Guicciardini 9, 21100 Varese, Italy; tzmario@inwind.it (M.T.-Z.); lucaspirito@hotmail.it (L.S.); alkis.karligkiotis@gmail.com (A.K.); paolo.castelnuovo@me.com (P.C.); 3Service de Neuroradiologie, AP-HP, Hôpital Lariboisière, 75010 Paris, France; jean-pierre.guichard@aphp.fr

**Keywords:** petrous apex, cholesterol granuloma, temporal bone, sphenoid sinus, trans-nasal approach

## Abstract

Objective: Petrous apex cholesterol granuloma (PACG) is a lesion that can give rise to different symptoms, and correlations with etiopathology are ambiguous. The aim of this study is to analyze the association between PACG symptoms and radiological findings at presentation, in order to establish a reproduceable pre-operative radiological evaluation and guide the surgical indication. Methods: PACG patients were collected in two tertiary care hospitals. All cases underwent CT/MRI to evaluate the cyst localization and erosion of surrounding structures. Typical and atypical symptoms were then analyzed and compared to radiologic findings established in accordance with the literature. Results: Twenty-nine patients were recruited; the most common symptoms were headache (69%), diplopia (20.7%) and fainting (24.1%), an atypical clinical manifestation related to jugular tubercle involvement. Significant associations between symptoms and radiologic findings were noted in terms of headache and temporal lobe compression (*p* = 0.04), fainting and jugular tubercle erosion (*p* < 0.001), vestibular symptoms and internal auditory canal erosion (*p* = 0.02), facial paresthesia and Meckel’s cave compression (*p* = 0.03), diplopia and Dorello canal involvement (*p* = 0.001), and tinnitus and cochlear basal turn erosion (*p* < 0.001). All patients were treated via an endoscopic–endonasal approach, in which extension was tailored to each case. At a median follow-up of 46 months, 93.1% of patients experienced resolution of symptoms. Conclusions: This clinico-radiological series demonstrates associations between symptoms and anatomical subsites involved with PACG. Hence, it may guide the surgeon at the time of surgical decision, since it asserts that typical and atypical symptoms are actually related to PACG.

## 1. Introduction

Petrous apex cholesterol granuloma (PACG) is one of the most frequent benign lesions of the temporal bone and typically arises from a well-pneumatized petrous apex [1]. Histologically, PACG is an intraosseous cyst contained in a thick fibrous capsule, with no true epithelial lining, which is filled with granulation tissue and a dark viscous fluid [2]. Traditionally, PACG is believed to arise as an inflammatory reaction to cholesterol crystals released from breakdown products in blood, although its origin is still disputed [2]. In particular, two major theories have been proposed: (1) blockage of the drainage pathways within the petrous apex resulting in a vacuum and consequent hemorrhage into this air space; (2) exposure of bone marrow due to pneumatized petrous apex that leads to hemorrhage, foreign-body reaction, and consequent obstruction of air cell flow [3]. PACG characteristically appears on CT as a well-defined, bony expansive mass, with remodeling and erosion of the surrounding bony structures; by MRI, PACG presents a unique hyperintense signal on T1-T2 weighted images with no enhancement after gadolinium injection.

From a clinical point of view, PACG is often asymptomatic or with vague symptoms and may be discovered incidentally; however, the correct etiopathological correlation between symptoms referred and the radiologic lesion is essential since a surgical approach is generally considered only in symptomatic cases [4]. In addition, even if a large PACG that causes neurologic dysfunction may be a reasonable candidate for surgical therapy, weighing the expected benefits with the potential surgical risks can be challenging in many patients, rendering specific studies of the structures involved of utmost importance. [4] Specifically, when symptoms occur due to encroachment on neurovascular structures, the most typical include unilateral headache, hearing loss, fullness, tinnitus, facial paresthesia, vestibular impairment, and diplopia. Nevertheless, other non-typical symptoms have been described as a consequence of PACG compression/invasion of the carotid artery/jugular bulb region and inferior/superior petrosal sinuses, even if no specific correlation between radiologic findings and symptoms referred has been described thus far [2,3,5].

The aim of this study is to retrospectively analyze the association between PACG typical/atypical symptoms and radiological findings at presentation, in order to establish a guide to surgical indication; in addition, the overall success rate of a personalized trans-nasal approach is reported.

## 2. Materials and Methods

This is a multicentric retrospective observational study conducted at two tertiary care referral centers that enrolled patients affected with PACG and treated accordingly. To be included, all patients needed to have available pre-operative radiological imaging studies (CT and MRI) that demonstrated the presence of PACG. In particular, CT scans revealed the presence of a bony expansive mass into the petrous apex, with possible erosion of the surrounding bony structures; conversely, on MRI, PACG presented a hyperintense T1-T2 signal with no gadolinium enhancement. 

Imaging data from each patient were examined by an expert radiologist and an ENT specialist focusing attention not only on the location of the cyst, but also on the adjacent structures involved. Considering the available literature on symptoms and hypothesized regions of radiological involvement, we established a reproduceable pre-operative radiologic map to better correlate typical symptoms, such as headache, hearing loss, vestibular symptoms, tinnitus, otitis media, diplopia, and facial paresthesia, with PACG. The following regions were considered: (1) temporal lobe/posterior fossa compression (Figure 1); (2) Meckel’s cave compression (Figure 1); (3) Eustachian tube compression/erosion (Figure 2); (4) internal auditory canal erosion (Figure 3); (5) cochlear basal turn erosion (Figure 4); (6) Dorello canal involvement (Figure 5); (7) jugular tubercle erosion. The latter parameter was introduced to evaluate its possible correlation with an atypical clinical presentation, namely fainting. To standardize all these radiological examinations, the jugular tubercle was defined in axial sections as the bony convexity medial to the jugular foramen just above the hypoglossal canal; in coronal sections, it is the bony part directly above the hypoglossal canal (Figure 6).

All patients were treated via an endoscopic-assisted trans-nasal surgical approach, which was defined as trans-sphenoidal, trans-pterygoid, or trans-clival in relation to the extension of the approach. Informed consent was obtained from each patient for treatment and use of de-identified clinical data for study purposes; the study, which was conducted according to the ethical standards established in the 1964 Declaration of Helsinki and revised in 2011, was approved by the Hopital Lariboisiere review board (CNIL no. 2226104) and by the Insubria Board of Ethics (approval number 0033025/2015).

Data on age, sex, laterality of the lesion, symptoms referred, surgical techniques, follow-up, and recurrence were collected.

### Statistical Analysis

Statistical analysis was performed with SPSS version 24 (IBM Corp. in Armonk, NY, USA). Data are reported as median ± interquartile range. A generalized linear model (GLM) was used to assess treatment outcomes. Fixed and random factors were age, gender, laterality of the lesion, and pre-surgical symptoms. Fisher’s exact test was used to evaluate the association of pre-surgical symptoms and radiologic findings. The level of statistical significance was set at *p* < 0.05.

## 3. Results

A total of 29 patients affected with PACG were recruited for this study: 17 females (58.6%) and 12 males (41.4%). The median age was 35 years (range 30–45.5); 19 cases presented a right-side lesion (65.5%) and 10 a left-side one (34.5%); 24 cases were treated with primary surgery (82.8%), whereas in five cases (17.2%), revision surgery was performed. 

Pre-surgical symptoms of PACG patients are described in Table 1; there was no significant association (*p* > 0.05) between symptoms referred and lesion laterality, age, or sex.

Among all included cases, the radiologic involvement of each structure was as follows: jugular tubercle erosion in eleven cases (37.9%), internal auditory canal erosion in six cases (20.7%), cochlear basal turn erosion in one case (3.4%), Eustachian tube compression/erosion in nine cases (31%), temporal lobe/posterior fossa compression in 16 cases (55.2%), Meckel’s cave compression in 10 cases (34.5%) and Dorello’s canal involvement in 11 cases (37.9%). It is noticeable that the combination of the involved structures on each patient was variable and related to its size.

When the association between symptoms referred and radiological region involved by the PACG was analyzed (Table 2), a significant association was noted between headache and temporal lobe/posterior fossa compression (*p* = 0.04, Cramer’s V 0.444), fainting and jugular tubercle erosion (*p* < 0.001, Cramer’s V 0.722), vestibular symptoms and internal auditory canal erosion (*p* = 0.02, Cramer’s V 0.536), facial paresthesia and Meckel’s cave compression (*p* = 0.03, Cramer’s V 0.468), diplopia and Dorello canal involvement (*p* = 0.001, Cramer’s V 0.653), and tinnitus and cochlear basal turn erosion (*p* < 0.001, Cramer’s V 1). The latter association should be carefully considered due to the presence of only one patient who experienced tinnitus with concomitant erosion of the cochlea basal turn. No associations were observed between headache and Meckel’s cave compression (*p* = 0.107), hearing loss and cochlear basal turn erosion (*p* = 0.103) or internal auditory canal erosion (*p* = 0.1), tinnitus and internal auditory canal erosion (*p* = 0.2), and otitis media and Eustachian tube compression (*p* = 0.287).

Considering surgical approaches, the majority of cases were treated with a trans-sphenoidal approach (16/29, 55.2%), eight cases with a trans-pterygoid (27.6%), and five cases with a trans-clival approach (17.2%). At a median follow-up of 46 months (range 15–111.5), 93.1% (27/29) of patients experienced resolution of symptoms, while two cases (6.9%) had no improvement in headache, as the pre-surgical symptom referred. However, due to the pathologic resolution demonstrated at follow-up CT, no further surgical interventions were performed. Statistical analysis showed no significant association between surgical success rate and pre-surgical symptoms, laterality, sex, or age (*p* > 0.05).

The peri-operative complications included two cases of nasal synechiae, one case of Hadad flap donor site infection, and one case of cerebrospinal fluid leak intraoperative occurrence that was successfully managed. 

## 4. Discussion

PACG is a benign lesion often characterized by vague symptoms that may delay its diagnosis. When present, several typical symptoms have been described to be related to this lesion, but no direct and definite association with radiologic PACG findings have been demonstrated, leaving its interpretation to clinical experience [3]. The aim of the study was therefore to establish if any a causality between ambiguous symptoms and PACG topography was present.

The main outcome of our analysis is that the typical symptoms of PACG have a significant association with specific radiologic findings (Table 2), with the exception of otitis media and hearing loss; in addition, we described fainting as an atypical clinical finding, highlighting its association with jugular tubercle erosion caused by the PACG. To the best of our knowledge, this is the first study that describes fainting as a related symptom to PACG and is the first time that a significant association has been demonstrated between PACG symptoms and specific anatomical structures analyzed radiologically.

The symptoms of PACG generally drive its treatment: in particular, for the most part, a surgical approach should be offered to symptomatic cases, even if expansion of lesion to critical structures has been described as a possible indication for surgery as well [3]. As a consequence, correct identification of PACG-specific symptoms is important. Over the past years, typical clinical manifestations have been described such as vertigo, diplopia, facial paresthesia/paresis, otitis media, tinnitus, hearing loss, and headache [6,7]. Even if the latter two are generally described as being the most frequent, correct interpretation of headache should be carefully considered since multiple causes of cephalic pain are possible, and its specific link with the pathologic process should be carefully considered [2,8]. It has been pointed out that headache is common, and efforts should be made to determine if it is directly related to PACG [2,4].

To solve this problem and establish a correct correlation between clinical symptoms and radiological PACG expansion, efforts have been made to standardize a pre-surgical radiologic check-list of the structures involved. In particular, considering our results, the following structures should be pre-operatively analyzed: temporal lobe/posterior fossa, Meckel’s cave, jugular tubercle, Dorello canal, Eustachian tube, internal auditory canal and cochlear basal turn. Moreover, depending on the localization of the cyst in PACG, in 2002, Mosnier et al. [9] described three possible patterns of clinical presentations related to involvement of three different adjacent structures: (1) otologic and vestibular symptoms, mainly determined by involvement of the internal auditory canal, sporadically associated with erosion of the basal cochlear turn; (2) temporal headache and facial pain attributable to compression of the dura of the temporal/posterior fossa, with possible involvement of trigeminal and abducent nerves, indicating the compression of Meckel’s cave region, and facial nerve involvement in case of extensive lesions; (3) serous otitis media caused by a compression of the eustachian tube [9]. That report represented the first attempt to associate clinical and radiological findings in PACG; nevertheless, its main limitations were the small sample size and the absence of statistical analyses between the pattern described and radiological findings.

In our study, we overcame this limitation and demonstrated the presence of a statistically significant association between most symptoms reported and radiological structures involved (Table 2) that enables drawing of a definitive conclusion in case of ambiguous PACG symptoms. Significant associations were noted between headache and temporal lobe/posterior fossa compression (*p* = 0.04, Cramer’s V 0.444), vestibular symptoms and internal auditory canal erosion (*p* = 0.02, Cramer’s V 0.536), facial paresthesia and Meckel’s cave compression (*p* = 0.03, Cramer’s V 0.468), diplopia and Dorello canal involvement (*p* = 0.001, Cramer’s V 0.653), and tinnitus and cochlear basal turn erosion (*p* < 0.001, Cramer’s V 1). In addition, differently from previous reports [9], we found no specific pattern of symptoms that presented independently of PACG size (*p* > 0.05) and localization. Symptoms associated with PACG are mainly related to peri-lesional inflammation rather than its size such that when an endoscopic trans-nasal PACG marsupialization is performed, the lesion is not removed, but it reduces its inflammation and related symptoms.

Taking into consideration the possible evolution of the lesion over the jugular tubercle, in the past few years, we noted that several patients in our cohort presented fainting as one of the presenting symptoms. In particular, by radiological imaging, we found that fainting occurred in cases of jugular tubercle erosion (*p* < 0.001, Cramer’s V 0.722), which presented the highest Cramer’s value in our series, (Table 2) thus empowering the role of this association. This symptom was previously described but not discussed, leaving its interpretation open to debate [10]. Nevertheless, taking into account the anatomy of the jugular tubercle, the 9th, 10th, and 11th cranial nerves are located above this anatomical structure, with the glossopharyngeal nerve being the more medial and superior, and consequently, we speculate that it is more likely to be involved in PACG [11]. The involvement of the 9th nerve may lead to glossopharyngeal irritation that can cause bradycardia and loss of sympathetic tone and therefore syncopal episodes without a necessarily associated pain syndrome [12,13,14]. Due to its challenging association with PACG, in the past decades, its correlation with these lesions may have been underestimated, although its identification and association with PACG is of pivotal importance since it can support the need for an appropriate surgical treatment.

Considering the benign nature of the lesion, the milestone of PACG treatment is surgical drainage and aeration of the cavity, which can be performed through open surgical (e.g., middle cranial fossa and trans-temporal approaches) or endoscopic endonasal approaches (using either a medial, medial with carotid lateralization, or infrapetrous transpterygoid approach) [1,15]. Due to the characteristic growth pattern of PACG, compared to open approaches, a medial endoscopic trans-nasal approach, in properly selected cases, is associated with similar outcomes and less peri-operative morbidity, and it is more commonly used to create a natural drainage pathway into the sinonasal cavities [1,6]. As demonstrated by a recent meta-analysis by Tabet et al. [6], endoscopic endonasal approaches should be favored over the open ones not only for the absence of an external scar, but also for the better hearing improvement and lower rates of complications. Various endoscopic techniques have been described in the attempt to maintain patency of the drainage pathway due to the circumferentially de-epithelialized pathway created by drilling the clival bone [16]. Specifically, given the experience on the use of a mucosal flap in frontal sinus surgery, the application of pedicled flaps (e.g., Hadad nasoseptal flap) in PACG treatment has been proposed, and it is associated with controversial outcomes [6,10,15,17]. In addition, the application of stents can be considered for selected cases, even if their real impact on the final success rate has not been demonstrated [6].

Overall, recurrence of symptoms in our series was seen in 6.9% of cases, which is somewhat lower than that reported in the literature of 0–60%, supporting the role of a personalized endoscopic-assisted trans-nasal approach in the management of properly selected cases of PACG [3].

The main strength of this study is the demonstration of a significant association between PACG symptoms referred and anatomic structures of patients recruited in two tertiary care hospitals; in case of rare or ambiguous symptoms, this anatomo-clinical association might help the surgeon to guide PACG management. Nevertheless, the retrospective nature of this study should be noted as its main weakness.

## 5. Conclusions

PACG is a benign lesion whose management is based on referred symptoms. As demonstrated by this study, the significant association between typical and atypical symptoms, such as fainting and radiological findings, is of pivotal importance since it can be used to correlate clinical symptoms with PACG, helping the surgeon in its management. However, further prospective studies are needed to confirm our preliminary results.

## Figures and Tables

**Figure 1 jcm-11-04297-f001:**
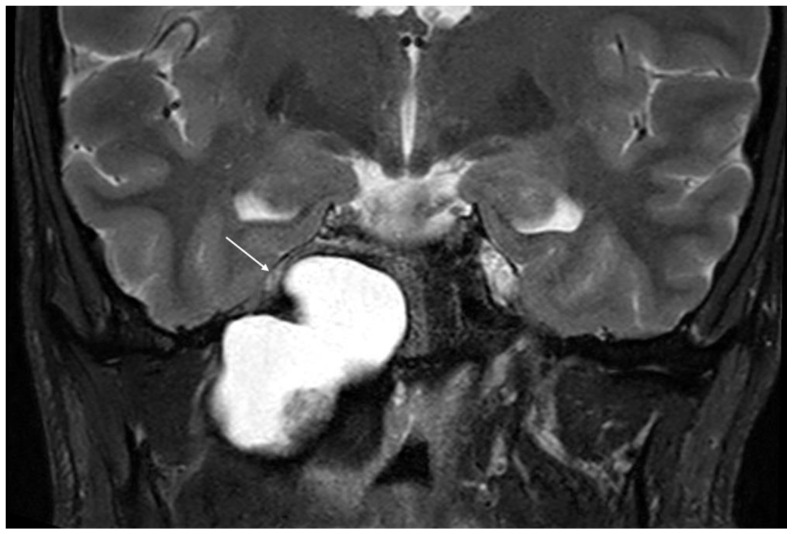
Coronal T2-weighted MRI demonstrating compression of the temporal lobe and Meckel’s cave region (white arrow) by cholesterol granuloma of the petrous apex.

**Figure 2 jcm-11-04297-f002:**
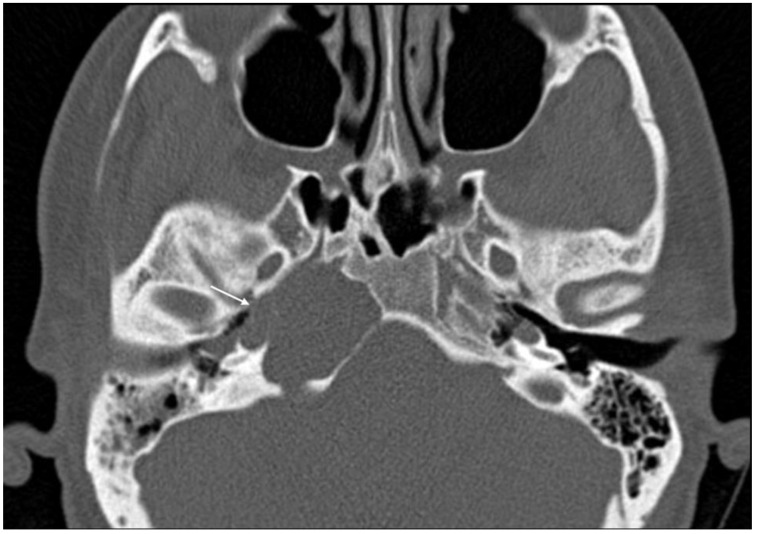
Axial CT scan showing compression of the Eustachian tube by a cholesterol granuloma (white arrow).

**Figure 3 jcm-11-04297-f003:**
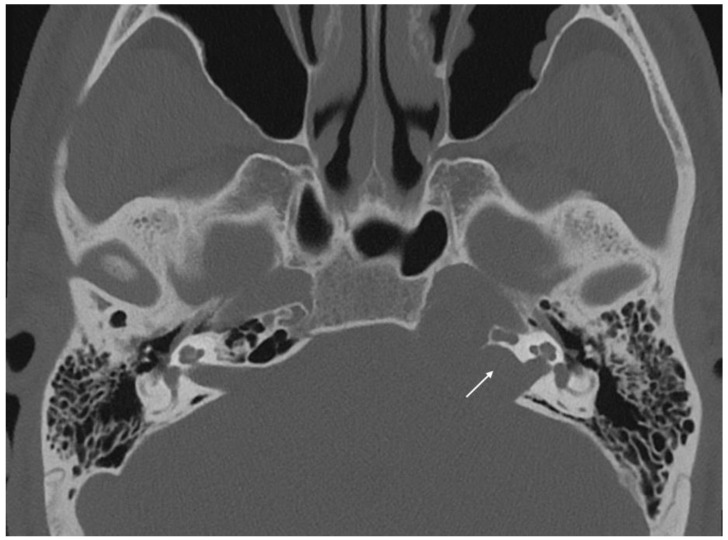
Axial CT scan showing erosion of the anterior wall of internal auditory canal (white arrow).

**Figure 4 jcm-11-04297-f004:**
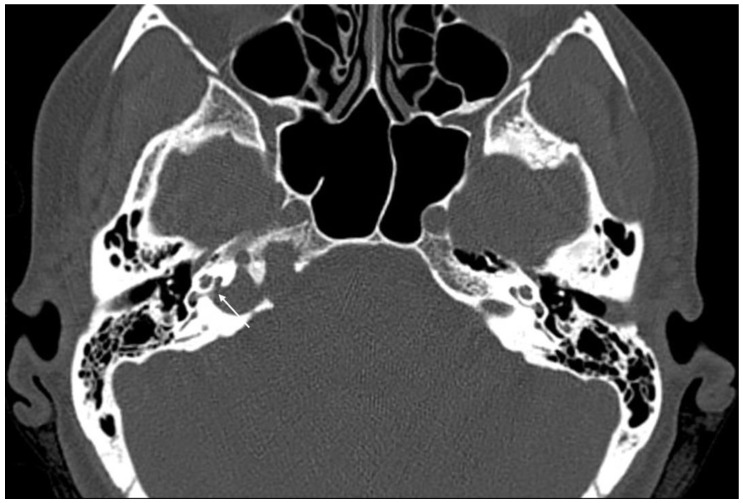
Axial CT scan showing erosion of the right cochlear basal turn (white arrow) by a cholesterol granuloma.

**Figure 5 jcm-11-04297-f005:**
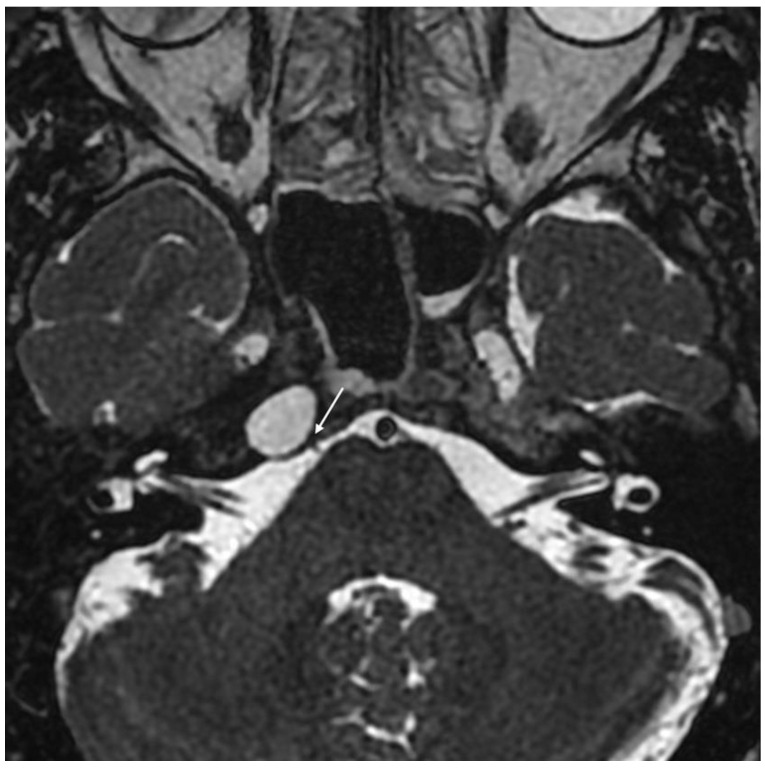
Axial T2-weighted MRI showing indirect compression of Dorello’s canal (white arrow) by a cholesterol granuloma of the petrous apex.

**Figure 6 jcm-11-04297-f006:**
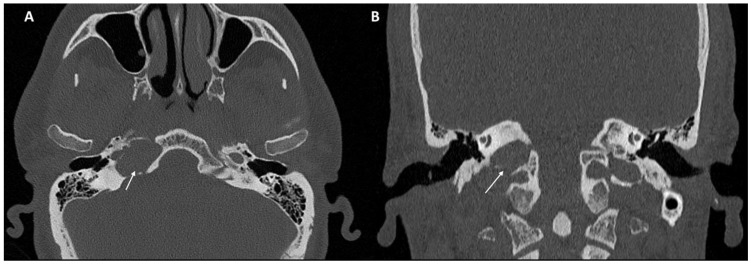
CT scan in the axial (**A**) and coronal (**B**) planes of the jugular tubercle (white arrow) defined as the bony convexity medial to the jugular foramen just above the hypoglossal canal (in axial sections) or the bony part just above the hypoglossal canal (in coronal sections).

**Table 1 jcm-11-04297-t001:** Pre-surgical symptoms of patients affected with cholesterol granuloma of the apex petrous. In some patients, more than one symptom was referred.

Pre-Surgical Symptom	No. N = 29	%
Headache	20	69%
Fainting	7	24.1%
Diplopia	6	20.7%
Otitis Media	5	17.2%
Vestibular symptoms	4	13.8%
Facial paresthesia	3	10.3%
Hearing loss	3	10.3%
Tinnitus	1	3.4%
Asymptomatic	1	3.4%

**Table 2 jcm-11-04297-t002:** Association between referred pre-surgical symptoms and radiologic structures involved with Fisher’s exact test and Cramer’s V index; significant results are shown in bold.

**Pre-Surgical Symptom**	**Radiologic Structures Involved**	**Fisher’s Exact Test**	**Cramer’s V**
Headache	Temporal lobe/posterior fossa compression	***p* = 0.04**	0.444
	Meckel’s cave compression	*p* = 0.107	-
Fainting	Jugular tubercule erosion	***p* < 0.001**	0.722
Diplopia	Dorello canal compression	***p* = 0.001**	0.653
Otitis Media	Eustachian tube compression	*p* = 0.287	-
Vestibular symptoms	Internal auditory canal erosion	***p* = 0.02**	0.536
Facial paresthesia	Meckel’s cave compression	***p* = 0.03**	0.468
Hearing loss	Cochlear basal turn erosion	*p* = 0.103	-
	Internal auditory canal erosion	*p* = 0.1	-
Tinnitus	Cochlear basal turn erosion	***p* < 0.001 ***	1.00
	Internal auditory canal erosion	*p* = 0.2	-

* only one patient of our cohort experienced tinnitus.

## Data Availability

Not applicable.

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
