# Peer review of "Typical and Atypical Symptoms of Petrous Apex Cholesterol Granuloma: Association with Radiological Findings"

_jcm, 2022, doi:10.3390/jcm11154297_

Round 1
Reviewer 1 Report
This is a very well written retrospective case series evaluating the association between radiographic findings and patient symptoms in petrous apex cholesterol granulomas. The manuscript reads smoothly from start to finish and the conclusion is well supported by the methods and results.
- Would be potentially beneficial for the authors to provide a suggested preoperative 'radiological checklist' of findings to review when evaluating these cases for surgery.
- Would recommend rephrasing the multiple uses of 'non-univocal' to improve readability (e.g. ambiguous)
- Consider stating that 100% of patients were treated with an endoscopic endonasal surgical approach? Currently the authors separate the surgical approaches out into transsphenoid/transclival/transpterygoid, but I think more importantly than the specific endonasal corridor taken, the more important finding here is that endoscopic endonasal approaches are a great management option for petrous apex cholesterol granulomas
Author Response
Thank you very much for your comments and suggestions.
Reviewer: This is a very well written retrospective case series evaluating the association between radiographic findings and patient symptoms in petrous apex cholesterol granulomas. The manuscript reads smoothly from start to finish and the conclusion is well supported by the methods and results.
- Would be potentially beneficial for the authors to provide a suggested preoperative 'radiological checklist' of findings to review when evaluating these cases for surgery.
Answer: Thank you, following your suggestion, we have added such information (line 223-225)
“In particular, considering our results the following structures should be pre-operatively analyzed: temporal lobe/posterior fossa, Meckel’s cave, jugular tubercle, Dorello’s canal, Eustachian tube, internal auditory canal and cochlear basal turn.”
Reviewer: - Would recommend rephrasing the multiple uses of 'non-univocal' to improve readability (e.g. ambiguous)
Answer: following you comment, we have modified the manuscript. Thanks.
Reviewer: - Consider stating that 100% of patients were treated with an endoscopic endonasal surgical approach? Currently the authors separate the surgical approaches out into transsphenoid/transclival/transpterygoid, but I think more importantly than the specific endonasal corridor taken, the more important finding here is that endoscopic endonasal approaches are a great management option for petrous apex cholesterol granulomas
Answer: yes you are right, we have modified the manuscript underling that all patients were treated with an endoscopic endonasal approach (see line 31-32, abstract, and 128-130). However, understanding your precious comment, the extension of the surgical approach we performed on our patients is very important since such information is substantially absent in the literature and could help future analysis on the best surgical approach for this benign pathology.
Reviewer 2 Report
The correlation between symptoms and imaging findings is clinically important.
The authors analyzed the correlation between symptoms and imaging findings in petrous apex cholesterol granuloma.
Please correct the following;
The number of radiologic structures involved was not described. Which findings were seen in how many cases? Also, were there cases in which multiple imaging findings were included? If so, it is considered necessary to describe them.
The authors have not described the results of the association between lesion size and MRI intensity and symptoms.Please describe these.
Author Response
Dear reviewer, thank you very much for you comments and kind suggestions. Here you can find our point-by-point reply.
- Reviewer: The correlation between symptoms and imaging findings is clinically important. The authors analyzed the correlation between symptoms and imaging findings in petrous apex cholesterol granuloma. Please correct the following; The number of radiologic structures involved was not described. Which findings were seen in how many cases? Also, were there cases in which multiple imaging findings were included? If so, it is considered necessary to describe them.
Answer: thank you very much for this precious comment. As suggested we have added the required information (line 158-163). The presence of multiple involved structures is highly variable (more than 18 combinations) and it could make confusion to the reader. However, we understand the importance of your message so that we have added a specific phrase on this topic
“Among all the included cases, the radiologic involvement of each structure was at follow: jugular tubercle erosion in 11 cases (37.9%), internal auditory canal erosion in 6 cases (20.7%), cochlear basal turn erosion in 1 case (3.4%), Eustachian tube compression in 9 cases (31%), temporal lobe/posterior fossa compression in 16 cases (55.2%), Meckel’s cave compression in 10 cases (34.5%) and Dorello’s canal involvement in 11 cases (37.9%). It is noticeable that the combination of the involved structures on each patient was variable.”
- Reviewer: The authors have not described the results of the association between lesion size and MRI intensity and symptoms. Please describe these.
Answer: dear reviewer, thank you for your comment. The mechanism that triggers symptoms in PACG is known to be related to peri-lesional inflammation rather than its size. This is also sustained by the fact that an endoscopic endonasal marsupialization does not provide removal of the lesion, but it allows to create a natural drainage pathway of such lesion into the sino-nasal cavities. As a result, the PACG volume remains the same, but we stop the inflammation and the symptoms accordingly. We understand your point that highlights a very important issue so that we have added the absence of significancy between lesion size and symptoms occurred, and a specific phrase on this topic (line 248-252).
“In fact, symptoms associated to PACG are mainly related to peri-lesional inflammation rather than its size so that when an endoscopic trans-nasal PACG marsupialization is performed, the lesion is not removed but it allows to reduce its inflammation and related symptoms.”
Reviewer 3 Report
The authors present a very interesting study on the radiological findings and the correlation with symptoms in patients with cholesterol granuloma of petrous apex. Despite a small sample, they manage to find significant differences in several radiological findings related to symptoms. To highlight the relationship between erosion/compression of the jugular foramen and fainting.
Despite being a retrospective study, I consider it to be a very interesting article.
Author Response
Thank you very much
Round 2
Reviewer 2 Report
The authors have responded appropriately to my points and I believe that the paper has been further brushed up.